# Novel Role of CETP in Macrophages: Reduction of Mitochondrial Oxidants Production and Modulation of Cell Immune-Metabolic Profile

**DOI:** 10.3390/antiox11091734

**Published:** 2022-08-31

**Authors:** Gabriel G. Dorighello, Leandro H. P. Assis, Thiago Rentz, Joseane Morari, Monique F. M. Santana, Marisa Passarelli, Neale D. Ridgway, Anibal E. Vercesi, Helena C. F. Oliveira

**Affiliations:** 1Department of Pathology, Faculty of Medical Sciences, State University of Campinas, Campinas 13083-888, SP, Brazil; 2Department of Structural and Functional Biology, Institute of Biology, State University of Campinas, Campinas 13083-862, SP, Brazil; 3Obesity and Comorbidities Research Center, State University of Campinas, Campinas 13083-864, SP, Brazil; 4Laboratório de Lípides (LIM10), Hospital das Clínicas (HCFMUSP), Faculdade de Medicina da Universidade de São Paulo, São Paulo 05403-000, SP, Brazil; 5Programa de Pós-Graduação em Medicina, Universidade Nove de Julho, São Paulo 01525-000, SP, Brazil; 6Departments of Pediatrics, and Biochemistry and Molecular Biology, Atlantic Research Center, Dalhousie University, Halifax, NS B3H4R2, Canada

**Keywords:** CETP, macrophage, mitochondria, oxidants, inflammation, cholesterol

## Abstract

Plasma cholesteryl ester transfer protein (CETP) activity diminishes HDL-cholesterol levels and thus may increase atherosclerosis risk. Experimental evidence suggests CETP may also exhibit anti-inflammatory properties, but local tissue-specific functions of CETP have not yet been clarified. Since oxidative stress and inflammation are major features of atherogenesis, we investigated whether CETP modulates macrophage oxidant production, inflammatory and metabolic profiles. Comparing macrophages from CETP-expressing transgenic mice and non-expressing littermates, we observed that CETP expression reduced mitochondrial superoxide anion production and H_2_O_2_ release, increased maximal mitochondrial respiration rates, and induced elongation of the mitochondrial network and expression of fusion-related genes (mitofusin-2 and OPA1). The expression of pro-inflammatory genes and phagocytic activity were diminished in CETP-expressing macrophages. In addition, CETP-expressing macrophages had less unesterified cholesterol under basal conditions and after exposure to oxidized LDL, as well as increased HDL-mediated cholesterol efflux. CETP knockdown in human THP1 cells increased unesterified cholesterol and abolished the effects on mitofusin-2 and TNFα. In summary, the expression of CETP in macrophages modulates mitochondrial structure and function to promote an intracellular antioxidant state and oxidative metabolism, attenuation of pro-inflammatory gene expression, reduced cholesterol accumulation, and phagocytosis. These localized functions of CETP may be relevant for the prevention of atherosclerosis and other inflammatory diseases.

## 1. Introduction

The cholesteryl ester transfer protein (CETP) was first described as a plasma protein that transfers cholesteryl ester from high-density lipoprotein (HDL) to very-low-density lipoprotein (VLDL) and low-density lipoprotein (LDL) in exchange for triglycerides [1]. Despite its effects on reducing HDL-cholesterol, the atherogenic role of CETP is still not firmly consolidated [2,3]. Human trials using CETP inhibitors were interrupted prematurely due to increased mortality or lack of benefits [2], although more recently, the use of the CETP inhibitor Anacetrapib showed significant reductions in cardiovascular events [4].

CETP may also indirectly modulate the cell cholesterol content. Sera from rabbits treated with a CETP inhibitor increased cholesterol efflux from Fu5AH hepatoma cells by 50% compared to control sera [5]. In contrast, plasma or HDL from CETP-expressing transgenic mice stimulated the cholesterol efflux from fibroblasts [6] and cholesteryl ester uptake by SRB1-expressing ovary cells [7]. In the case of macrophages, a pivotal cell type implicated in atherogenesis, conflicting results were reported. HDL or plasma from patients with high CETP activity increased cholesterol efflux from mice peritoneal macrophage [8] and THP-1 human monocytic cells [9]. However, HDL from patients with CETP deficiency also promoted higher cholesterol efflux from mice peritoneal macrophage and THP-1 cells [10]. Thus, the role of plasma or HDL-associated CETP in cholesterol accumulation and efflux is not well established. On the other hand, it has not been determined whether CETP expression by macrophages has a direct effect on cholesterol homeostasis.

Experimental and clinical studies suggest that CETP may have a novel anti-inflammatory function [11,12,13]. Transgenic mice expressing physiological concentrations of CETP exhibited reduced plasma inflammatory cytokines and lower mortality when exposed to bacterial lipopolysaccharide (LPS) or after polymicrobial sepsis induced by cecal ligation and puncture [11,13]. In addition, patients’ plasma CETP levels were correlated with sepsis survival [14], supporting findings of increased deaths by infections and sepsis in the CETP inhibitor Torcetrapib trial [15]. However, recent studies in mice and humans reported opposite results [16,17]. Apart from the model differences, in vivo, it is quite difficult to discriminate between the role of CETP and HDL concentrations in the susceptibility or protection against acute inflammation.

Inflammatory responses and oxidant production are interconnected events during both macrophage phenotype polarization [16] and the development of atherosclerosis [17]. The bactericidal and phagocytic activities of macrophages rely on NADPH-oxidase 2 (NOX2)-derived reactive oxygen species (ROS). Additionally, the production of mitochondrial ROS plays an important role in inflammation signaling during the immune response [18,19], and reduced mitochondrial ROS due to UCP2 activity decreases the inflammatory response [20,21].

Since CETP plays a relevant role in both atherosclerosis and inflammation, and mitochondrial ROS are connected with inflammatory signaling, we initiated a study to investigate whether CETP expression in macrophages modulates mitochondrial redox and bioenergetics functions and macrophage inflammatory profile, as well as its impact on cholesterol accumulation and phagocytic activity.

## 2. Materials and Methods

### 2.1. Animals

All experimental protocols were approved by the State University of Campinas (UNICAMP) Committee for Ethics in Animal Experimentation (CEUA/UNICAMP, protocol number 4746-1/2017) and are in accordance with the Ethical Principles of the National Council for the Control of Animal Experimentation (CONCEA) and the ARRIVE research guidelines for the use of Laboratory Animals (https://arriveguidelines.org, accessed on 28 July 2022) Mice were maintained in a temperature-controlled conventional room (22 ± 1 °C) on a 12 h light/dark cycle, 15 cycles of air change per hour and free access to filtered water and standard laboratory rodent chow diet (Nuvital CR1, Colombo, Brazil). Four or five mice were housed per cage. Both male and female mice aged 4–6 months were used as a source of cells for the experiments. Hemizygous CETP transgenic (Tg) mice expressing a human natural promoter-driven CETP transgene (line 5203, The Jackson Laboratory, Bar Harbor, ME, USA, RRID:IMSR_JAX:003904) were maintained in the Department of Structural and Functional Biology of the State University of Campinas (UNICAMP) since 1996, and cross-bred with C57BL6/JUnib (Multidisciplinary Center for Biological Research, CEMIB/UNICAMP) to generate CETP-Tg and non-Tg (control) littermates. Additional experiments were performed with mice overexpressing simian CETP cDNA controlled by metallothionein promoter (Tg [CETP] UCTP20Pnu/J) (The Jackson Laboratory, RRID:IMSR_JAX:001929) purchased in 2009 and maintained as described above. CETP genotyping was performed by polymerase chain-reaction (PCR) using tail tip DNA samples according to The Jackson Laboratory protocols. CETP genotyping and mRNA expression in peritoneal macrophages (PM) and bone marrow derived macrophages (BMDM) are shown in Appendix A.

### 2.2. Peritoneal Macrophages

Mice anesthetized with isoflurane (Isoforine, Cristália, Águas de Lindóia, Brazil) were euthanized by decapitation and the abdominal outer skin layer was carefully removed. Six mL of ice-cold phosphate buffered saline solution (PBS) was injected into the peritoneum and approximately 5 mL of cell-containing fluid was aspirated from the peritoneal cavity. The cell containing fluid was centrifuged at 400× *g*, 4 °C for 5 min. Supernatant was discarded and the cell pellet was gently resuspended in Roswell Park Memorial Institute (RPMI) 1640 medium (Vitrocell, Campinas, Brazil) supplemented with 10% (*v*/*v*) fetal bovine serum (Vitrocell). Cell counting was performed using a Muse^®^ Cell Analyzer (Millipore, Burlington, MA, USA). Peritoneal cells were diluted to 1 × 10^6^ cells/mL and seeded in a 96-well plate (100 µL/well). Two hours after incubation, cells were washed to remove non-adherent cells. After overnight incubation at 37 °C and 5% (*v/v*) CO_2_, these resting macrophages were ready for experimentation. Thioglycolate elicited peritoneal macrophages were obtained four days after an intraperitoneal injection of 1 mL sterile thioglycolate solution (3% *v/v*). Macrophages were isolated as described above, diluted to 1 × 10^6^ cells/mL, and seeded in 24-well (500 µL/well) or 96-well plates (100 µL/well). Two hours after incubation, cells were washed to remove non-adherent cells.

### 2.3. Bone Marrow Derived Macrophages (BMDM)

Bone marrow was aseptically flushed from femurs with Dulbecco’s Modified Eagle Medium (DMEM) containing high glucose (4.5 g/L glucose) (Vitrocell) and 10% (*v/v*) fetal bovine serum (Vitrocell). Cells were collected by centrifugation at 1000 × *g* (room temperature) for 5 min, and resuspended in 2 mL of Red Blood Cell Lysing Buffer (Sigma-Aldrich, St. Louis, MA, USA) for 5 min to lyse red blood cells and preserve nucleated white blood cells. Two ml of sterile PBS was added to neutralize the lysis and the suspension was centrifuged for 5 min at 1000× *g* (room temperature). The supernatant was discarded and cells were resuspended in DMEM supplemented with 15% (*v/v*) L929 cell-conditioned medium [22], 10% (*v/v*) fetal bovine serum and 1% (*w/v*) penicillin/streptomycin (Vitrocell). Cells were immediately cultured in complete medium for 7 days to induce differentiation into BMDM. On the fourth day, non-adherent cells were eliminated by changing one-half of the culture medium. On the eighth day, cells were detached using Accutase Cell Detachment Solution (Sigma-Aldrich), counted and replated for further experiments.

### 2.4. Oxidant Generation

To measure oxidant generation, macrophages were seeded on a black 96 well plate with μClear bottom (Greiner Bio-one, Kremsmünster, Austria).

H_2_O_2_ release was quantified using Amplex^®^ red (Molecular Probes, Eugene, OR, USA). Experiments were done in 100 µL PBS buffer pH 7.4 containing glucose (11.1 mM), Amplex^®^ Red Reagent (25 µM), horseradish peroxidase (HRP; 0.2 U/mL) in the dark at 37 °C for 40 min. Fluorescence of the formed resorufin was measured at 530 nm (excitation) and 590 nm (emission) every 10 min in SpectraMax M3 (Molecular Devices, Sunnyvale, CA, USA). In one well per group, catalase (500 U/mL) was added to correct for background fluorescence. To eliminate the mitochondria contribution, cells were incubated with carbonyl cyanide-4-(trifluoromethoxy) phenylhydrazone (FCCP; 1 µM). Thus, the mitochondrial H_2_O_2_ release was calculated by the difference between the total H_2_O_2_ release and FCCP-incubated cells.

The total intracellular and mitochondrial superoxide anion were determined using the fluorescent indicators dihydroethidium (DHE) and MitoSOX^®^ Red mitochondrial superoxide indicator, respectively (ThermoFisher Scientific, Waltham, MA, USA). The cells were treated with DHE (10 µM) or MitoSOX^®^ (2 µM), in PBS buffer pH 7.4, glucose (11.1 mM) in the dark at 37 °C for 30 min. Cells were then washed twice with PBS and total fluorescence quantified using a Spectramax-M3 multi-mode microplate reader (Molecular Devices). DHE fluorescence was monitored at 518 nm (excitation) and 605 nm (emission). MitoSOX fluorescence was monitored at 510 nm (excitation) and 580 nm (emission). Fluorescence was corrected by background fluorescence in wells without the probes.

To measure global oxidant levels in macrophages, 2′,7′-dichlorodihydrofluorescein diacetate (H_2_DCFDA) was used. When attacked by any oxygen or nitrogen reactive species, the probe is converted to a fluorescent 2’,7’-diclorofluorescein (DCF) product. Cells were treated with H_2_DCFDA (10 µM) (Thermo Fisher Scientific) in PBS buffer pH 7.4, glucose (11.1 mM) in the dark at 37 °C for 30 min. Cells were washed twice with PBS and DCF fluorescence was monitored using a Spectramax-M3 (Molecular Devices) multi-mode microplate reader at 492 nm (excitation) and 517 nm (emission). Fluorescence was corrected by the background fluorescence in wells without H_2_DCFDA.

### 2.5. Oxygen Consumption Rates (OCR) and Extracellular Acidification Rates (ECAR)

Macrophages were seeded at 1 × 10^5^ cells/well in 24 well plates designed for the Seahorse^®^ XF24 Extracellular Flux Analyzer (Agilent, Santa Clara, CA, USA). The dissolved oxygen concentration in the extracellular medium was evaluated, providing the cellular oxygen consumption rate (OCR) and the changes in the pH reflecting the rate of glycolysis. Twenty-four hours after cell seeding, the medium was changed and cells were treated as described below. One hour before the assay, cells were washed with RPMI medium without fetal bovine serum, sodium bicarbonate and phenol red, at 37 °C in a CO_2_-free atmosphere. Then, different drugs that modulate mitochondrial respiration rates were sequentially injected into the medium: oligomycin 1 µM (ATP synthase inhibitor), FCCP 1.25 µM (mitochondrial uncoupler), and a mix of antimycin A 1 µM (complex 3 inhibitor) and rotenone 1 µM (complex 1 inhibitor). The OCR measured in each condition were calculated as follows: 1. Non-mitochondrial oxygen consumption: The minimum rate measurement after rotenone and antimycin injection; 2. Basal respiration: (last rate measurement before the oligomycin injection) minus (non-mitochondrial oxygen consumption rate); 3. Maximal respiration: (maximal rate measurement after FCCP injection) minus (non-mitochondrial oxygen consumption rate); 4. Proton leak: (minimal rate measurement after oligomycin injection) minus (non-mitochondrial oxygen consumption rate); 5. ATP-linked respiration: (last rate measurement before the oligomycin injection) minus (minimal rate measurement after oligomycin injection); 6. Spare respiratory capacity: (maximal respiration) minus (basal respiration). The medium acidification rates calculations were: 1. Glycolysis: Last rate measurement prior to oligomycin injection; 2. Glycolytic reserve: (maximum measurement rate after oligomycin injection) minus (last rate measurement prior to oligomycin injection); 3. Glycolysis Capacity: (Glycolysis) plus (Glycolytic reserve).

### 2.6. Crystal Violet DNA Staining

Cells were washed with PBS and fixed with 100 µL/well of paraformaldehyde 4% (*w/v*) for 5 min at room temperature. Next, cells were washed and incubated with 100 µL/well crystal violet (0.05% *w/v*) diluted in distilled water for 10 min at room temperature. Cells were then washed thoroughly with water, dried and re-suspended with 100 μL/well of acetic acid (10% *v/v*). The absorbance was measured at 590 nm using a SpectraMax M3 multi-mode microplate reader (Molecular Devices).

### 2.7. Mitochondrial Network Analysis

Peritoneal macrophages were seeded on a black 96-well plate with µClear bottom (Greiner Bio-one). Cells were incubated with 200 ηM MitoTracker^®^ Deep Red FM (ThermoFisher Scientific) and 2 µg/mL Hoechst 33342 (ThermoFisher Scientific) diluted in RPMI-1640 medium (Vitrocell) without fetal bovine serum at 37 °C, 5% (*v/v*) CO_2_ for 45 min. Cells were washed with PBS and imaged using an ImageXpress Micro Confocal High-Content Imaging System (Molecular Devices). Automated analyses were performed as described previously [22]. Mitochondrial morphological analyses involved calculating the mitochondrial aspect ratio (elongation) as the ratio between its length and breadth. The form factor (branching) was calculated as (perimeter^2^)/(4π*area) [23]. Representative images were adjusted regarding background, contrast and brightness using the software Fiji [24].

### 2.8. Gene Expression by Real Time PCR

Total RNA was extracted using RNeasy kit (Qiagen, Venlo, The Netherlands). One µg of purified RNA was used to synthesize cDNA using a High-Capacity cDNA Reverse Transcription Kit (Applied Biosystems, Forster City, CA, USA). Real time PCR was performed using a Step-One Real-time PCR System (Applied Biosystems). Primers (Appendix A) were designed and tested against the *Mus musculus* genome (GenBank). The relative quantities of the target transcripts were calculated from duplicate samples using the ΔΔCT method [25]. Gene expression data were normalized by beta actin (ACTB) or 36b4 gene expression.

### 2.9. Cytokine Secretion Assay

Macrophage were seeded at 500,000 cells/well in a 24-well plate and the culture medium was collected after 24 h incubation at 37 °C and 5% (*v/v*) CO_2_. Interleukin-1β was quantified using Mouse Il-1β Uncoated ELISA kit (ThermoFisher Scientific) and the interleukin-10 using Mouse IL-10 Quantikine ELISA Kit (R&D Systems, Minneapolis, MN, USA), according to the manufacturer’s protocols. For normalization, the cells were harvested and incubated with 100 µL RIPA buffer containing cOmplete Protease Inhibitor Cocktail (Roche, Basel, Switzerland) at 4 °C for 10 min. The cellular lysate was centrifuged at 18,000× *g*, 4 °C for 15 min. The protein content was determined in the supernatant using BCA Protein Assay (ThermoFisher Scientific).

### 2.10. Phagocytosis Activity

Macrophages were seeded in 96-well flat-bottom tissue culture plate at 100,000 cells/well. Then, 10 µL of neutral-red stained zymosan (1 × 10^8^ particles/mL) were added to each well and the cells were incubated at 37 °C, 5% (*v/v*) CO_2_ for 30 min. Next, the medium was removed and the cells were fixed with Baker’s solution (4% (*w/v*) formaldehyde, 2% (*w/v*) sodium chloride, 1% (*w/v*) calcium acetate) at 37 °C for 30 min. The macrophages were washed twice with PBS and the neutral-red stain was solubilized with 0.1 mL of acidified alcohol solution: 10% (*v/v*) acetic acid, 40% (*v/v*) ethanol in distilled water. After 30 min, the absorbance was monitored at 550 nm in the SpectraMax M3 (Molecular Devices) as previously described [26]. The phagocytosis index was expressed related to crystal violet DNA staining in control wells.

### 2.11. Cell Cholesterol Content

Macrophages were incubated at 37 °C with 50 µg/mL of human oxidized LDL (oxLDL, Alfa Aesar, Ward Hill, MA, USA) or PBS for 24 h in RPMI-1640 medium supplemented with 1% (*v/v*) fetal bovine serum (Vitrocell). The cells were washed twice with PBS buffer and lysed with RIPA buffer. Ten µL of each sample and 40 µL of catalase (12.5 U/mL) was mixed in a black 96-well plate, followed by incubation at 37 °C for 15 min in order to eliminate any peroxides present in reagents or samples [27]. Next, 50 μL of cholesterol oxidase/Amplex Red Reagent (0.1 M potassium phosphate buffer, pH 7.4, 50 mM NaCl, 5 mM cholic acid, 0.1% (*v/v*) Triton X-100, 2 U/mL cholesterol oxidase, 0.2 U/mL cholesterol esterase, 2 U/mL HRP, and 100 µM Amplex Red) was added to each well. Unesterified cholesterol content was measured with the same reagents but in the absence of the cholesterol esterase. The plate was incubated at 37 °C for an additional 15 min and fluorescence was read at 530 nm (excitation) and 590 nm (emission). The cholesterol content was normalized by total protein in the lysate using the BCA Protein Assay (ThermoFisher Scientific).

### 2.12. Unesterified Cholesterol Determination by Filipin Staining

Macrophages seeded in a black 96-well plate with μClear bottom (Greiner Bio-one) were washed twice with PBS buffer, fixed with paraformaldehyde 4% (*w/v*) and incubated with filipin (100 µg/mL) for 1 h. Cells were then washed twice with PBS buffer and fluorescence was quantified using a SpectraMax M3 multi-mode microplate reader (Molecular Devices) at 360 nm (excitation) and 480 nm (emission). The background fluorescence was corrected in wells without filipin.

### 2.13. Cholesterol Efflux Assay

Mouse resting peritoneal macrophages and BMDM were treated with acetylated LDL (50 µg/mL) and 3 µCi/mL of ^14^C-cholesterol (Amersham Biosciences, Little Chalfont, UK) for 24 h. After two washes with PBS containing fatty acid free albumin (FAFA), macrophages were incubated for 48 h in DMEM (Gibco) containing 1 mg/mL FAFA to allow equilibration of intracellular cholesterol pools. After washing the cells, human HDL (50 µg/mL) was added as a cholesterol acceptor for 24 h. Medium was collected and centrifuged to remove cell debris and cells were extracted with hexane/isopropanol (3:2; *v/v*). After solvent evaporation, radioactivity was determined and ^14^C-cholesterol efflux estimated as [^14^C-cholesterol in the medium/(^14^C-cholesterol in the medium + ^14^C-cholesterol in cells) × 100]. Values were subtracted from those obtained in incubations with DMEM/FAFA alone (basal efflux) in order to determine the HDL-specific cholesterol efflux activity.

### 2.14. CETP Knockdown by siRNA

THP-1 human monocytic cells was cultured in RPMI-1640 medium plus 10% (*v/v*) fetal bovine serum at 37 °C and 5% (*v/v*) CO_2_. THP-1 monocytes were differentiated into macrophages using 200 ηM of phorbol 12-myristate 13-acetate (PMA) for 72 h. Macrophages were transfected with a siRNA pool specific for hCETP (L-009485-00-0005) or a non-targeting siRNA (D-001810-0X) using DharmaFECT 4 reagent (T-2004-01) (Dharmacon™, Horizon Discovery Ltd., Cambridge, UK) for 72 h according to the manufacturer’s instructions. The amount of siRNA was 1 ρmol/well (96-well plate) for filipin staining or 5 ρmol/well (24-well plate) for gene expression assay.

### 2.15. Statistical Data Analyses

The results are presented as the means ± standard error of the mean (SEM) and individual values. The comparisons between the groups were analyzed by unpaired Student’s t test using Prism 8.0 software (GraphPad, San Diego, CA, USA) and the level of significance was set at *p* ≤ 0.05.

## 3. Results

### 3.1. CETP Expression in Macrophages Decreases Mitochondrial Oxidant Production

In order to determine whether CETP expression affects cell redox state, we measured oxidant production in resting resident and thioglycolate-elicited peritoneal macrophages. Mitochondrial superoxide generation measured with the MitoSOX probe was significantly reduced (12%) in resting macrophages from CETP transgenic mice (Figure 1A). The rate of H_2_O_2_ release measured using the Amplex Red probe was also decreased (23%) in resting macrophages from the CETP transgenic mice (Figure 1B,C). Since resident resting macrophages present an immature phenotype [28], we further studied cells obtained three days after an intraperitoneal injection of thioglycolate, which induces a mild sterile inflammation. Global cell oxidants (probed with H_2_DCF-DA) (Figure 1D) and total superoxide anion (probed with DHE) (Figure 1E) production in thioglycolate-elicited peritoneal macrophages were not modulated by CETP expression. However, CETP expression reduced mitochondrial-derived superoxide anion (probed with MitoSOX) by 28% compared to controls (Figure 1F). CETP expression also reduced total (17%) and mitochondrial (17%) H_2_O_2_ release in thioglycolate-elicited peritoneal macrophages (Figure 1G,H). The generation of non-mitochondrial H_2_O_2_ was not affected by macrophage CETP expression (Figure 1I).

These results were further confirmed in peritoneal macrophages from an independent line of transgenic mice overexpressing simian CETP (Appendix A). We also induced a pro-inflammatory polarization by treating these thioglycolate-elicited peritoneal macrophages with lipopolysaccharide (LPS) from *Escherichia coli* or tumor necrosis factor α (TNFα). Neither treatment changed the results observed in non-activated peritoneal macrophages (Appendix A), that is, CETP expression decreases mitochondrial oxidant production.

### 3.2. CETP Expression in Macrophages Increases Mitochondrial Maximal Respiration

We measured macrophage bioenergetics profile by determining the oxygen consumption rates (OCR) and extracellular acidification rates (ECAR) (Figure 2A,B). Macrophages from CETP-Tg mice displayed a 23% and 65% increase in the maximal respiration rate and in the spare respiratory capacity, respectively (Figure 2E,F). Respiration rates in other states (basal, associated with ATP production, proton leak, and non-mitochondrial) were similar in macrophages from CETP-Tg and non-expressing mice (Figure 2C,D,G,H). These results were confirmed in peritoneal macrophages from an independent line of transgenic mice expressing simian CETP, which showed increased mitochondrial maximal respiration (33%) and spare respiratory capacity (66%) as compared to cells from non-transgenic mice (Appendix A).

Cell glycolytic metabolism was determined by measuring ECAR. CETP expression did not change basal glycolysis and glycolytic capacity (Figure 2I,K). However, after stimulating glycolysis with oligomycin, the CETP expressing macrophages exhibited a higher glycolytic reserve (glycolytic capacity minus basal glycolysis) (Figure 2J).

### 3.3. CETP Expression in Macrophages Increases Mitochondrial Network Elongation

We investigated the mitochondrial network morphology, an indication of mitochondrial dynamics (fission and fusion balance). Using MitoTracker dye and an automated identification of mitochondrial network parameters in a micro-confocal high-content imaging system, we determined the mitochondrial aspect ratio, which indicates the degree of mitochondrial elongation, and the form factor, the degree of mitochondrial branching. Indeed, the mitochondrial elongation parameter was increased in CETP expressing macrophages, while branching was similar in both groups (Figure 3A–C). Corroborating these results, CETP expressing macrophages also increased the expression of mitochondrial dynamin like GTPase genes mitofusin-2 (MFN2) and OPA1 (Figure 3D), which encode proteins involved in mitochondrial fusion process [29].

### 3.4. Effects of CETP Expression on Inflammation Related Genes, Cytokine Secretion and Phagocytosis

Since cell redox changes are part of inflammation signaling pathways, we evaluated the effect of CETP expression in macrophages on inflammatory gene expression. We found that the expression of TNF-α, IL-6 and iNOS was reduced in peritoneal macrophages and bone marrow derived macrophages (BMDM) expressing CETP (Figure 4A,D). However, mRNA and secreted protein for the classical inflammatory cytokine IL-1β were increased in CETP peritoneal macrophages (Figure 4A,C), but not in BMDM (Figure 4D,F). The mRNA and secreted protein for the anti-inflammatory IL-10 were not modulated by CETP expression in peritoneal macrophages (Figure 4A,B) and BMDM (Figure 4D,E).

To determine the impact of CETP expression on macrophage immune function, we measured phagocytosis of zymosan particles stained with neutral red. CETP-expressing peritoneal macrophages (Figure 5A) and BMDM (Figure 5B) both presented with reduced phagocytosis activity.

### 3.5. Macrophages Expressing CETP Have Reduced Unesterified Cholesterol Content

Macrophages with a M2 anti-inflammatory phenotype are resistant to lipid accumulation [30]. Since CETP expression caused a mild anti-inflammatory phenotype (Figure 4), we investigated if CETP expression could modulate macrophage cholesterol homeostasis. First, we analyzed cholesterol metabolism and transport genes in peritoneal macrophage (Figure 6A). Compared to non-Tg controls, we observed reduced CD36 gene expression (fatty acids and oxidized LDL uptake) but increased expression of the LDL receptor (LDL uptake), HMG-CoA reductase (cholesterol synthesis) and ABCA1 (cholesterol efflux) genes (Figure 6A). CETP-expressing BMDM had reduced gene expression for CD36 but increased ABCG1 gene expression (Figure 6B). These profiles are consistent with CETP promoting reduced cholesterol content in macrophages.

Next, we measured cell cholesterol content under basal conditions and after incubation with oxidized LDL. We found no significant changes in the total cholesterol content (Figure 7A,D). However, both PM and BMDM expressing CETP showed less unesterified cholesterol in both conditions (Figure 7B,E). Filipin staining of membrane cholesterol confirmed a reduction in unesterified cholesterol in CETP-expressing PM (Figure 7C). In addition, HDL-mediated cholesterol efflux was markedly enhanced (2.8×) in CETP expressing PM (Figure 7F).

In order to confirm that reduced unesterified cholesterol content was dependent on the CETP expression, we knocked down CETP gene expression in human monocytic THP1 cells using siRNA (Figure 8). A 43% reduction in CETP mRNA (Figure 8A) resulted in a 32% increase in unesterified cholesterol content (Figure 8B). Compared to the results of CETP expression on mitochondrial fusion, inflammatory and cholesterol regulatory genes (Figure 3, Figure 4 and Figure 6), CETP knockdown in TPH-1 cells abolished the upregulation of mitofusin-2 and the downregulation of TNFα expression, but was not enough to reverse the downregulation of CD36 mRNA (Figure 8A).

## 4. Discussion

Local oxidative stress is a key event in immune system cells such as macrophages, which have a central role in atherogenesis and other inflammatory diseases. CETP expression or activity is suggested to be a regulator of acute inflammation and atherosclerosis, although the often-conflicting results from experimental and clinical studies have failed to elucidate whether the effects of CETP are direct or indirect (through lowering HDL) in theses disease contexts.

Here, we evaluated the features of macrophages obtained from mice expressing physiological (similar to humans) plasma levels of CETP and mild changes in HDL levels as compared with control littermates not expressing CETP. Considering that (i) in vitro, CETP favors the M2 macrophage anti-inflammatory phenotype after inflammatory stimulation [12], and (ii) the classical inflammation signaling pathway mediated by toll like receptors (TLR) involves mitochondria-derived oxidants mediators [19,31], we postulated that CETP could have an anti-inflammatory role by influencing mitochondria bioenergetics and redox functions. Indeed, we demonstrated that CETP expression promoted a reduction in the mitochondrial superoxide production in resting, thioglycolate-elicited and LPS- or TNFα-stimulated peritoneal macrophages. In addition, mitochondrial H_2_O_2_ release was also decreased in CETP expressing macrophages. Elevated respiratory rates are associated with lower mitochondrial superoxide generation [32] and with anti-inflammatory macrophage phenotype [33]. Conversely, superoxide dismutase-deficient mice, which accumulate H_2_O_2_, had reduced mitochondrial spare respiratory capacity [34], indicating that the mitochondrial antioxidant capacity is associated with their respiratory capacity. Accordingly, we found that CETP-expressing macrophages exhibited increased mitochondrial maximal respiration rates and spare respiratory capacity. The increased maximal respiration rates can be attained by a higher efficiency of the respiratory chain and/or increased endogenous substrate uptake and metabolism [35]. The latter could also explain the increased glycolytic reserve found in the CETP-expressing macrophages. In agreement, female mice expressing simian CETP fed a high fat diet had improved exercise capacity due to increased muscle mitochondrial oxidation of malate/glutamate substrates [36].

We also evaluated whether the CETP expression could change the morphology of mitochondrial network that is tightly associated with the organelle functionality. Macrophages expressing CETP had a prominent elongated mitochondrial network, suggesting an organelle dynamic that favors mitochondrial fusion over fission. This interpretation is supported by the increased expression of MFN-2 and OPA1, major players in mitochondrial fusion, in CETP expressing macrophages. This finding may be associated with lower mitochondrial oxidant production, since fusion is inhibited whereas fission is stimulated by mitochondrial oxidants [37]. Furthermore, elongated mitochondria present a more efficient electron transport chain due to super-complex formation, which increases the respiratory capacity and reduces electron leak in immune cells [38].

Variations in the levels of cellular ROS contribute to inflammatory signaling pathways. Indeed, we found that CETP-expressing macrophages, which exhibit low mitochondrial oxidant production rates, showed downregulation of classical pro-inflammatory genes, such as iNOS, TNFα, and IL-6. Noteworthy, CD36, a member of the scavenger receptors family with a well characterized role in the uptake of oxLDL, was also downregulated in CETP-expressing macrophages. CD36 ligands have also been shown to promote sterile inflammation through assembly with TLR4 or TLR6 heterodimer [31,39]. In line with our results, a previous study showed that addition of exogenous CETP to cultured mouse peritoneal macrophages that do not express CETP had decreased LPS uptake, TLR4 expression, NF-𝜅B activation and release of IL-6 [11]. Thus, it seems that both endogenous CETP expression as well as CETP from exogenous sources (e.g., recombinant CETP or plasma with high CETP levels) can attenuate the inflammatory pattern of macrophages.

We also observed an increased expression and secretion of the pro-inflammatory cytokine IL-1β in CETP-expressing peritoneal macrophages, but not in BMDM. IL-1 gene expression is regulated at the transcriptional and post-transcriptional levels by a wide range of physical and chemical factors. One possible explanation for increased IL-1β secretion is the relationship between mitofusin-2, which was upregulated in CETP macrophages, and NLRP3. According to Ichinohe and collaborators, mitofusin-2 interaction with NLRP3 increases inflammasome activation, which then promotes IL-1β secretion in macrophages [40]. The upregulation of IL-1 expression in CETP-expressing macrophages is likely related to the thioglycolate stimulus, since IL-1β upregulation was observed only in PM but not in BMDM. Nonetheless, IL-1β expression and secretion is also a feature of the alternatively activated, anti-inflammatory M2b macrophage phenotype [41], showing the complexity and plasticity of macrophage polarization phenotypes. Reduced phagocytosis activity suggests a degree of immune response suppression in CETP-expressing macrophages.

The balance between cholesterol uptake and efflux in macrophages is a determinant of the foam cell formation, an early event in atherogenesis. Since CETP-expressing macrophages had decreased oxidant production and exhibited an attenuated inflammatory profile, we hypothesized that CETP expression in macrophages would also decrease cholesterol accumulation. In fact, CETP expressing macrophages exhibit lower concentrations of unesterified cholesterol both in basal conditions and after incubation with oxLDL in PM and BMDM. Lower CD36 expression likely contributed to the decreased uptake of oxLDL. In addition, HDL-mediated cholesterol efflux was markedly increased in resting PM, but not in BMDM. One plausible reason may be related to the cell differentiation processes. While PM are terminally differentiated primary cells, BMDM differentiate in vitro under artificially induced conditions that may or may not reproduce the complete features of PM. In accordance with decreased oxLDL uptake (in which CD36 play a major role) and increased cholesterol efflux (in which ABCA1/G1 play the major roles), we did observe a reduction in the unesterified cholesterol but not in total cholesterol content. This is explained by the well-known intracellular cholesterol homeostasis regulation that led to the upregulation of LDL receptor and HMG-CoA reductase expression (as shown in Figure 6).

The lower unesterified cholesterol content in macrophage membranes was confirmed to be dependent on CETP expression since knockdown of the CETP gene in human THP1 cells resulted in increased unesterified cholesterol content, as shown by filipin staining, widely used as a probe for sterol location in biological membranes. Membrane cholesterol accumulation in macrophage activates inflammation through TLR and p38 mitogen-activated protein kinase [42]. Conversely, macrophages with reduced membrane cholesterol display inflammatory signaling suppression [43,44]. Thus, reduced unesterified cholesterol in CETP-expressing macrophages may be associated with the attenuation of their inflammatory profile.

During atherosclerotic plaque progression, a vicious cycle between oxidative stress and inflammation is established [45]. Accordingly, aged LDL receptor knockout mice transplanted with bone marrow that overexpresses mitochondrial catalase to reduce mitochondrial H_2_O_2_ production, presented with less atherosclerosis compared to mice transplanted with wild-type bone marrow [46]. Thus, the observed CETP-dependent mitochondrial antioxidant effects and downregulation of pro-inflammatory genes in macrophages might be relevant to the initiation and progression of atherosclerosis.

Altogether, we showed that CETP expression in macrophages decreased mitochondrial oxidant production, increased mitochondrial respiratory capacity, induced mitochondrial network elongation, attenuated the expression of pro-inflammatory genes, and decreased unesterified cholesterol content and phagocytosis activity. These overall changes in the macrophage immune-metabolic profile reveal new local functions of CETP that might be relevant for atherosclerosis and other inflammatory contexts.

## Figures and Tables

**Figure 1 antioxidants-11-01734-f001:**
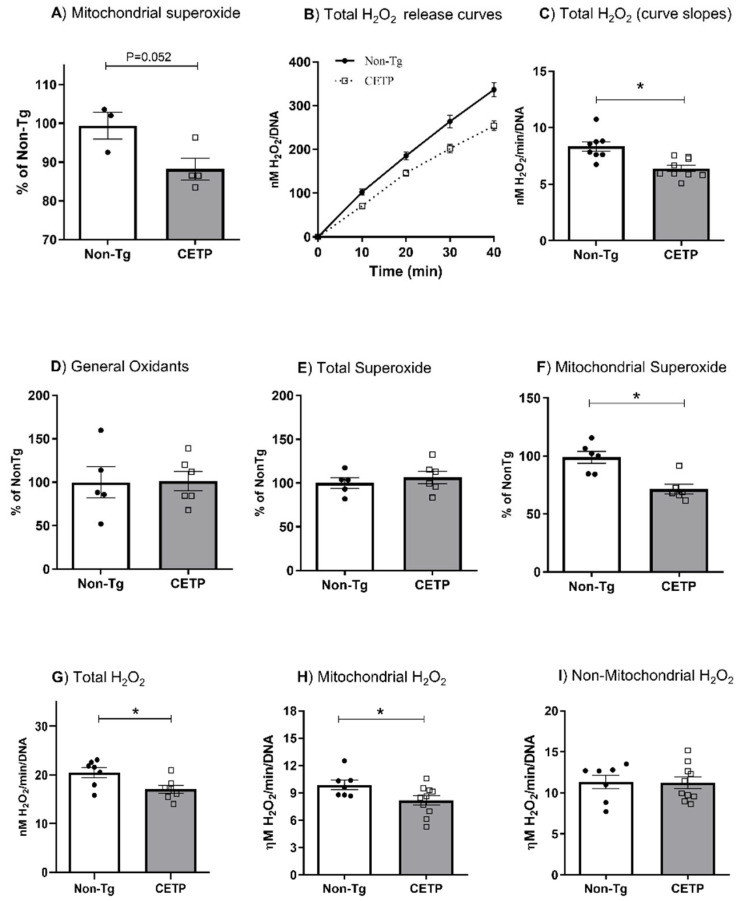
CETP expression in macrophages decreases mitochondrial oxidant production. Oxidant production in resting resident (**A**–**C**) and thioglycolate-elicited (**D**–**I**) peritoneal macrophages from human CETP transgenic and non-transgenic mice (Non-Tg). (**A**) Mitochondrial superoxide anion production; (**B**) H_2_O_2_ release curves; (**C**) H_2_O_2_ release rates (curve slopes from B); (**D**) global oxidant production; (**E**) total cell superoxide; (**F**) mitochondrial superoxide anion; (**G**) total H_2_O_2_; (**H**) mitochondrial H_2_O_2_ and (**I**) non-mitochondrial H_2_O_2_ release rates. Non-mitochondrial H_2_O_2_ contribution was determined in cells treated with carbonyl cyanide-p-trifluoromethoxyphenylhydrazone (FCCP) (1 µM) and mitochondrial derived H_2_O_2_ was calculated as (total - non-mitochondrial H_2_O_2_). * *p* < 0.05.

**Figure 2 antioxidants-11-01734-f002:**
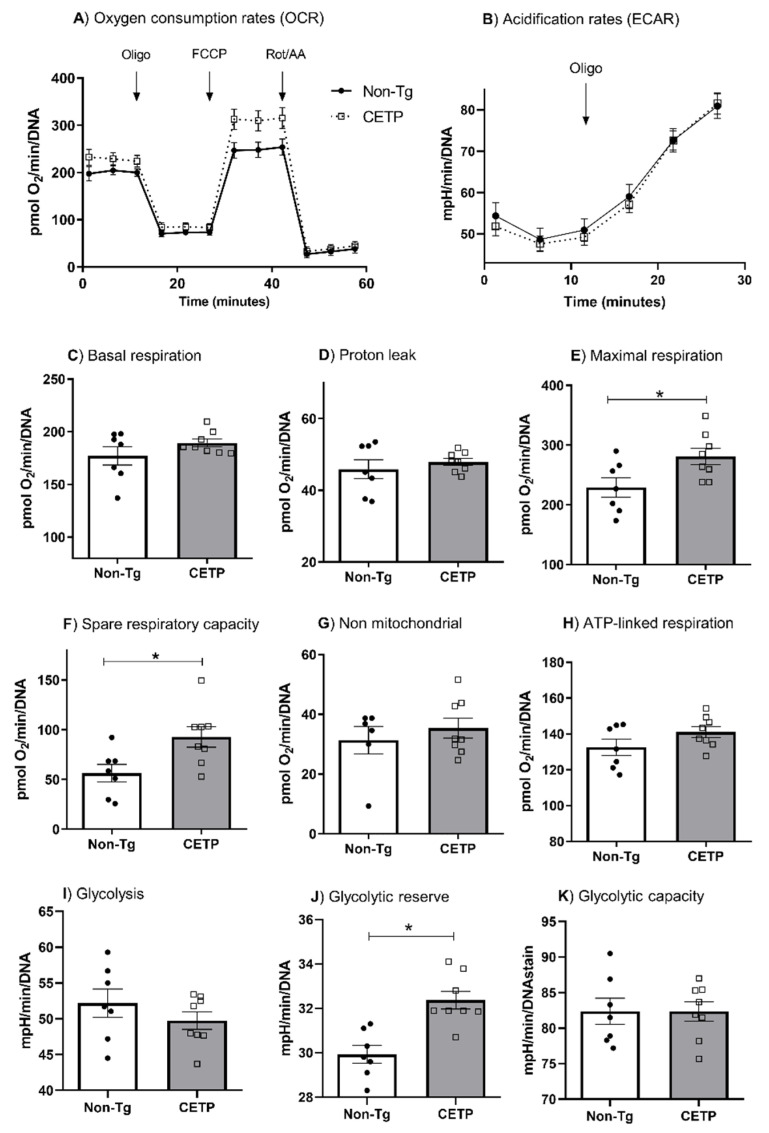
CETP expression in macrophages increases mitochondrial maximal respiration rate, spare respiratory capacity and glycolytic reserve. Oxygen consumption rates (OCR) and extracellular acidification rates (ECAR) were determined in thioglycolate-elicited peritoneal macrophages from human CETP transgenic mice and non-transgenic mice (Non-Tg). (**A**) Average curve of OCR with sequential additions of oligomycin (1 µM), FCCP (1.25 µM), and rotenone/antimycin A (1 µM); (**B**) Average curve of ECAR with oligomycin (1 µM); (**C**) basal respiration; (**D**) proton leak; (**E**) maximal respiration; (**F**) spare respiratory capacity; (**G**) non-mitochondrial respiration; (**H**) ATP-linked respiration; (**I**) glycolysis; (**J**) glycolytic reserve; (**K**) glycolysis capacity. * *p* < 0.05.

**Figure 3 antioxidants-11-01734-f003:**
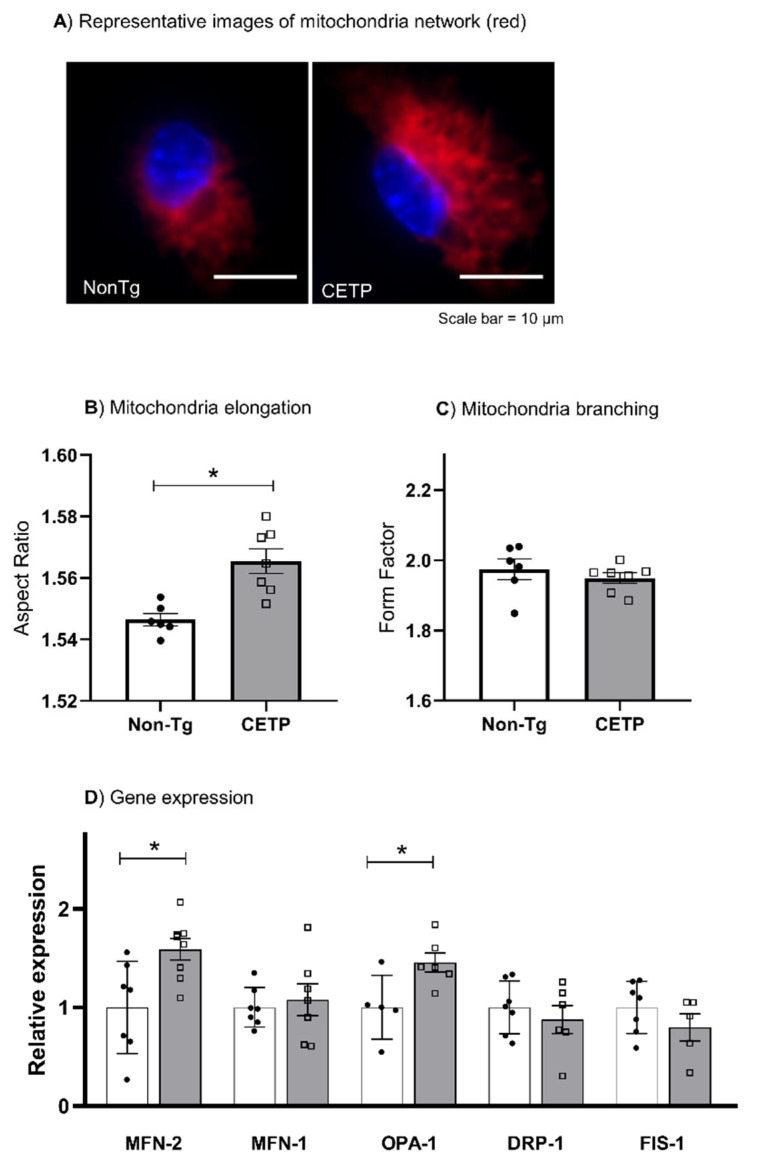
CETP expression in macrophages increases mitochondrial network elongation and the expression of fusion-related genes. Mitochondrial network morphology and gene expression were determined in thioglycolate-elicited peritoneal macrophages from human CETP-Tg and non-transgenic (Non-Tg) mice. (**A**) Representative images (60× magnification) showing mitochondria stained with MitoTracker (red) and nuclei stained with Hoechst 33342 (blue). (**B**) The aspect ratio represents the degree of mitochondrial elongation and (**C**) the form factor represents the extent of network branching degree. (**D**) mRNA expression of the fusion genes Mitofusin-2 (MFN-2), MFN-1, and OPA-1, and the fission genes dynamin-related protein 1 (DRP-1) and mitochondrial fission 1 protein (FIS-1). * *p* < 0.05.

**Figure 4 antioxidants-11-01734-f004:**
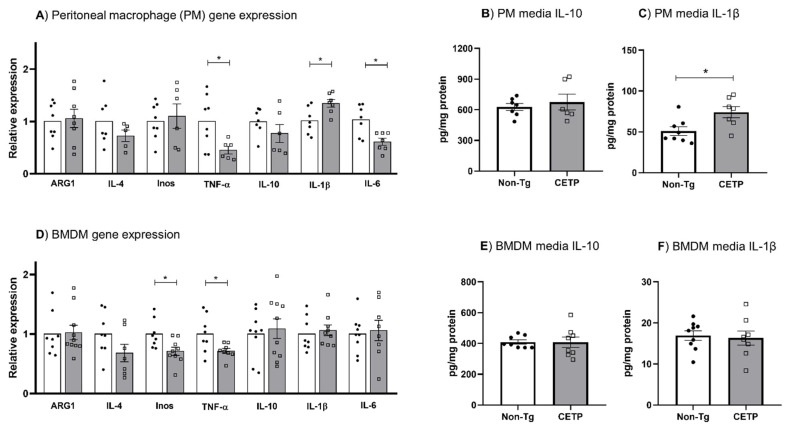
Effects of CETP expression in macrophages on inflammatory related genes and cytokines secretion. Gene expression (**A**,**D**) and IL-10 and IL-1β media secretion (**B**,**C**, and **E**,**F**) in thioglycolate-elicited peritoneal macrophage (**A**–**C**) and in bone marrow derived macrophage (BMDM) (**D**–**F**) from human CETP-Tg and non-transgenic mice (Non-Tg). * *p* < 0.05.

**Figure 5 antioxidants-11-01734-f005:**
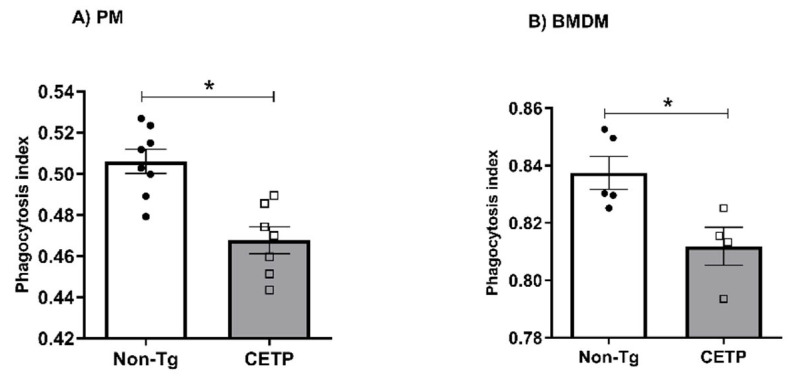
Macrophages expressing CETP have reduced phagocytosis. Phagocytic activity measured in thioglycolate-elicited peritoneal macrophage (PM) (**A**) and in bone marrow derived macrophage (BMDM) (**B**) from human CETP-Tg and non-transgenic mice (Non-Tg). * *p* < 0.05.

**Figure 6 antioxidants-11-01734-f006:**
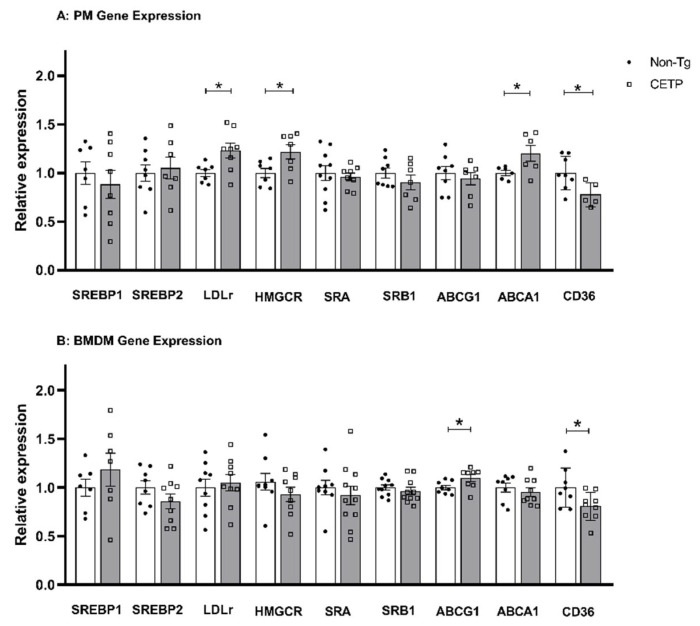
CETP expression in macrophages changes the expression of cholesterol homeostasis related genes. Relative gene expression levels were measured in thioglycolate-elicited peritoneal macrophages (PM) (**A**) and in bone marrow derived macrophages (BMDM) (**B**) from human CETP-Tg and non-transgenic mice (Non-Tg). * *p* < 0.05. Sterol response element binding protein 1 and 2 (SREBP1, SREBP2), LDL receptor (LDLr), 3-hydroxy-3-methyl-glutaryl-CoA reductase (HMGCR), scavenger receptor class A (SRA), scavenger receptor class B member 1 (SRB1), ATP-binding cassette transporter A1 and G1 (ABCA1, ABCG1), cluster determinant 36/fatty acid translocase (CD36).

**Figure 7 antioxidants-11-01734-f007:**
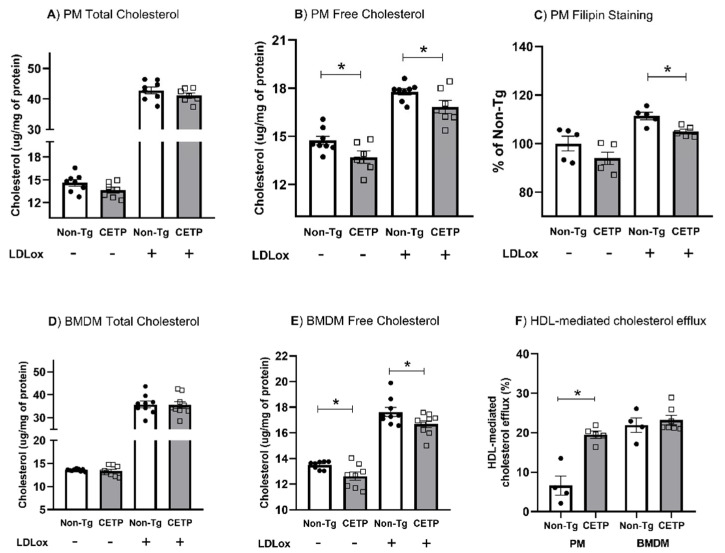
Macrophages expressing CETP present with reduced unesterified cholesterol content and increased cholesterol efflux. Cholesterol content before and after exposure to oxidized LDL (LDLox) were measured in thioglycolate-elicited peritoneal macrophage (PM, **A**,**B**) and in bone marrow derived macrophage (BMDM, **D**,**E**). Unesterified cholesterol was also measured by Filipin staining (**C**). HDL mediated cholesterol efflux was measured in resting PM and BMDM (**F**). See Material and Methods for details. * *p* < 0.05.

**Figure 8 antioxidants-11-01734-f008:**
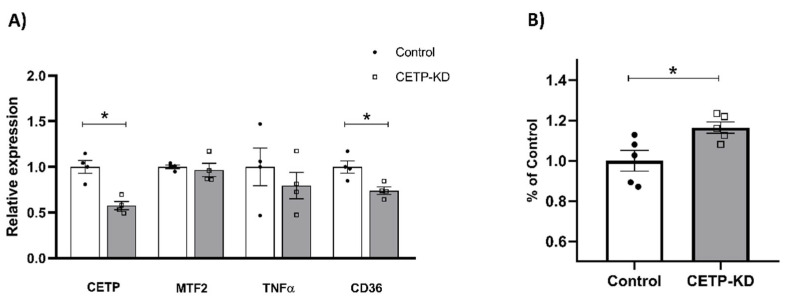
Knockdown of CETP in THP1 cells increased unesterified cholesterol content and did not affect mitofusin-2 and TNFα gene expression. THP1 human cells treated with siRNA targeted to CETP or control siRNA were evaluated for mRNA levels of key genes (**A**) and stained for membrane cholesterol with filipin (**B**). * *p* < 0.05.

## Data Availability

The data that support the findings of this study are available from the corresponding author upon reasonable request.

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
