# Peer review of "Novel Role of CETP in Macrophages: Reduction of Mitochondrial Oxidants Production and Modulation of Cell Immune-Metabolic Profile"

_antioxidants, 2022, doi:10.3390/antiox11091734_

Round 1
Reviewer 1 Report
In this study, the authors found that macrophages, obtained from human CETP transgenic mice, had antioxidant state and anti-inflammatory property, through modulating mitochondrial structure and function. Moreover, these macrophages reduced cholesterol accumulation and phagocytosis. This is an interesting study, the observed effect of CETP could be relevant for prevention of atherosclerosis and other inflammatory diseases. However, there are some concerns:
1) CETP expression (mRNA, endogenous) should be measured in the transgenic mouses microphages, even though both endogenous and exogenous CETP can attenuate the inflammatory pattern of macrophages.
2) Why promoting cholesterol efflux only observed in peritoneal macrophages but not bone marrow derived macrophages? This should be discussed.
3) All observed phenotypes could be related with less membrane free cholesterol accumulation. Thus, macrophage membrane free cholesterol needs to be measured.
Author Response
Please, see the attachment.

Reviewer 2 Report
The author provided evidence supporting that macrophages from transgenic mice expressing human CETP have a less atherosclerotic immune-metabolic profile compared to those from non-trangenic mice. Data from this manuscript are sufficient to support changes in cellular respiration, mitochondrial ROS production, and cholesterol metabolism in transgenic macrophages. However, alterations in pro-inflammatory cytokine (except IL-1b) and CD36 expressions were only examined at the RNA level. The author will need other methods (e.g., ELISA or Western Blot) to show the changes also occur at the protein level. Also, although peritoneal macrophages from transgenic mice showed an increase in cholesterol efflux, the total cholesterol content had no change. Please provide a possible explanation for this phenomenon in the discussion section. Besides, the author needs to clarify the full gene name of "MTF2". For readers' convenience, please use relative centrifugal force instead of rotations per minute in the methodology section.
Author Response
Please, see the attachment.

Round 2
Reviewer 1 Report
No more comments.